# No-Show in Medical Appointments with Machine Learning Techniques: A Systematic Literature Review

**Luiz Henrique Américo Salazar** [1], **Wemerson Delcio Parreira** [1,*], **Anita Maria da Rocha Fernandes** [1,*] and **Valderi Reis Quietinho Leithardt** [2,3]

1. Master Program in Applied Computer Science, School of Sea, Science and Technology, University of Vale do Itajaí, Itajaí 88302-901, Brazil
2. VALORIZA, Research Center for Endogenous Resources Valorization, Instituto Politécnico de Portalegre, 7300-555 Portalegre, Portugal
3. COPELABS, Universidade Lusófona de Humanidades e Tecnologias, 1749-024 Lisboa, Portugal
* Correspondence: parreira@univali.br (W.D.P.); anita.fernandes@univali.br (A.M.d.R.F.)

**Abstract:** No-show appointments in healthcare is a problem faced by medical centers around the world, and understanding the factors associated with no-show behavior is essential. In recent decades, artificial intelligence has taken place in the medical field and machine learning algorithms can now work as an efficient tool to understand the patients' behavior and to achieve better medical appointment allocation in scheduling systems. In this work, we provide a systematic literature review (SLR) of machine learning techniques applied to no-show appointments aiming at establishing the current state-of-the-art. Based on an SLR following the PRISMA procedure, 24 articles were found and analyzed, in which the characteristics of the database, algorithms and performance metrics of each study were synthesized. Results regarding which factors have a higher impact on missed appointment rates were analyzed too. The results indicate that the most appropriate algorithms for building the models are decision tree algorithms. Furthermore, the most significant determinants of no-show were related to the patient's age, whether the patient missed a previous appointment, and the distance between the appointment and the patient's scheduling.

**Keywords:** no-show; medical appointments; healthcare; artificial intelligence; data processing and management

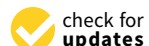



## 1. Introduction

Medical appointment no-shows are a problem in essentially all healthcare centers, and have a significant impact on revenues, costs and the use of resources. Also referred as missed appointments, the no-show states the patient's behavior who neither attend nor cancel their medical appointments. Long waiting lines for a particular medical resource, hindering health system access for other patients and financial loss are well-known consequences that the patients' absence from the scheduled appointments can cause.

Previous studies have looked at the economic consequences of a patient's no-show. Mesa et al. [1] pointed out that the cost of rescheduling an appointment is approximately 13 euros, and showed that in Spain, the cost generated by patient no-shows was around 3 million euros. To reduce these adverse effects, health centers have implemented various strategies, including overbooking [2–4] and reminders [5–7].

Despite the development of medical scheduling systems to achieve better appointment management, the high rate of missed appointments still remains a problem. With the advent of artificial intelligence (AI) in the last decades, machine learning (ML) algorithms can serve as efficient tools to understand the patients' behavior and also to achieve better appointment allocation based on predictive models. As shown by Chong et al. [8], there was a reduction from 19.3% to 15.9% in no-shows, by sending a reminder to 25% of the

patients who were indicated by the predictive model at greater risk of not attending the appointment.

There is growing interest from the healthcare community in understanding the issues involved in no-show behavior. However, identifying the features that most influence this behavior is still a challenging problem, given the variability of contexts in healthcare centers. Thus, by aggregating studies that focus on understanding the no-show behavior in medical appointments using machine learning techniques, it is possible to address the problem properly. Moreover, no updated systematic literature review exists in this field.

This paper addresses a systematic review to establish the state-of-the-art in machine learning techniques focused on no-show appointments. The review aims to identify the models that have been proposed and also which are the most widely used ML algorithms in the literature. Besides identifying each of the different ML techniques, various elements such as the characteristics of the database, algorithms employed, and the performance obtained are analyzed.

We adopt the systematic literature review following a procedure based on the Preferred Reporting Items for Systematic Reviews and Meta-Analyses (PRISMA) statement by [9], and organize the remainder of this paper as follows. In Section 2 we detail the search protocol, including the databases selected, search criteria, and the attributes extracted from each selected study. Next, in Section 3, the selected studies are grouped according to each research question, and the contributions of each one are highlighted. Finally, in Section 4 we discuss our findings and present the conclusions in Section 5.

## 2. Materials and Methods

First, based on the PRISMA procedure, we formulated the research questions, determined the keywords, and determined the inclusion and exclusion criteria. We selected databases in the fields of health, social sciences, and computer science, determined as the most relevant fields for assessing studies related to machine learning techniques applied to patient no-shows in medical attendances.

Based on the main goal of this systematic review, four research questions were developed. According to the healthcare area in which the machine learning techniques are applied, we observe different patterns and highlight some attributes that can contribute to a better understanding of the patient's no-show behavior. Dataset size and features explored, initial no-show rate, and machine learning algorithms trained and tested, are examples of different characteristics that can provide relevant information in this context. So, each question was focused on answering one of the following topics: data volume used to train the model, machine learning algorithms, most important features that influence the no-show behavior, and the impact of using machine learning in this field. All research questions are presented in Table 1.

**Table 1.** Research Questions.

| ID | Research Question |
|----|-------------------|
| Q1 | How much data is used to train machine learning models for no-show prediction? |
| Q2 | Which machine learning algorithms are used in the predictive models and which of these presented the best performance? |
| Q3 | What characteristics most influence patients not to attend a scheduled medical appointment? |
| Q4 | What is the no-show rate reduction in medical appointments that solutions developed with the help of machine learning techniques achieve? |

Due to the fact that several terms are not strict, a search expression was defined not only including the broad term "no-show", but also searching with terms such as "non-

attendance" and "absenteeism". The definition of the final search string is a result of several calibration searches realized beforehand. The detailed search string was then translated to the specific syntax of each database (see Table 2).

A database search was conducted in PubMed, Scopus, Web of Science, IEEE Xplore, and ScienceDirect screening for relevant articles. As the initial search results returned only articles from the last five years, we kept this time boundary in all database searches. Then, we selected articles published between 1 January 2017 and 1 January 2022 to capture the most recent research in machine learning applied to patient no-shows.

**Table 2.** Search strings.

| Database | Search Strings |
|---|---|
| Scopus | (TITLE-ABS-KEY(no-show) OR TITLE-ABS-KEY(absenteeism)) AND TITLE-ABS-KEY("machine learning") AND (TITLE-ABS-KEY(medical) OR TITLE-ABS-KEY(healthcare)) AND (TITLE-ABS-KEY(appointment) OR TITLE-ABS-KEY(consultation)) |
| IEEE Xplore | ((("All Metadata":"no-show") OR ("All Metadata":absenteeism)) AND ("All Metadata":"machine learning") AND (("All Metadata":medical) OR ("All Metadata":healthcare)) AND (("All Metadata":appointment) OR ("All Metadata":consultation))) |
| Science Direct | ("no-show" OR "absenteeism") AND ("machine learning") AND ("medical" OR "healthcare") AND ("appointment" OR "consultation") |
| PubMed | ((no-show) OR (absenteeism)) AND ("machine learning") AND ((medical) OR (healthcare)) AND ((appointment) OR (consultation)) |
| Web of Science | ((no-show) OR (absenteeism)) AND ("machine learning") AND ((medical) OR (healthcare)) AND ((appointment) OR (consultation)) |

The first step in study selection was formulating eligibility criteria, which we defined in terms of desirable characteristics of the study. The list of predefined inclusion and exclusion criteria is shown in Table 3. In this work, only academic research is being considered and proposals at a commercial level are not being taken into account.

**Table 3.** Inclusion and Exclusion Criteria.

| Category | Inclusion and Exclusion Criteria |
|---|---|
| Inclusion Criteria | The article must: Contain an abstract; Be written in English; Have been published between 1 January 2017 and 1 January 2022 |
| Exclusion Criteria | The article has a focus on: Medical attendance prediction in other fields than health appointments; Medical attendance prediction without the use of machine learning techniques; Literature reviews |

In addition to the inclusion and exclusion criteria, the studies were selected through the analysis of the (i) title; (ii) abstract and keywords; and (iii) conclusion. As a quality criterion, only works that contained substantial information were included, such as which machine learning techniques were used, which dataset attributes were available, and which algorithms and evaluation metrics were performed.

After applying the search expression in each database and subsequent initial article assessments, 63 articles were discovered. In the first stage, only the duplicated works were removed and only one version of them was kept for analysis. Most of the duplicate

articles were in the Scopus database and it was decided to keep the most up-to-date articles. Among these, two articles found in IEEE Xplore, seven in PubMed, and five in the ScienceDirect database were excluded from the Scopus results. An article was also removed from ScienceDirect results as it was already included in PubMed results.

In the second stage, the inclusion and exclusion criteria were applied and 24 articles were selected. The articles kept in the second stage were selected for a full reading. Table 4 shows the number of articles found and selected for each database in the respective steps mentioned. Figure 1 depicts the full selection process of articles.

**Table 4.** Number of articles discovered and selected in each stage.

| Database | Discovered | 1st Stage | 2nd Stage |
|---|---|---|---|
| IEEE Xplore | 3 | 3 | 3 |
| PubMed | 7 | 7 | 6 |
| Science Direct | 10 | 9 | 4 |
| Scopus | 30 | 15 | 10 |
| Web of Science | 13 | 2 | 1 |
| Total | 63 | 36 | 24 |

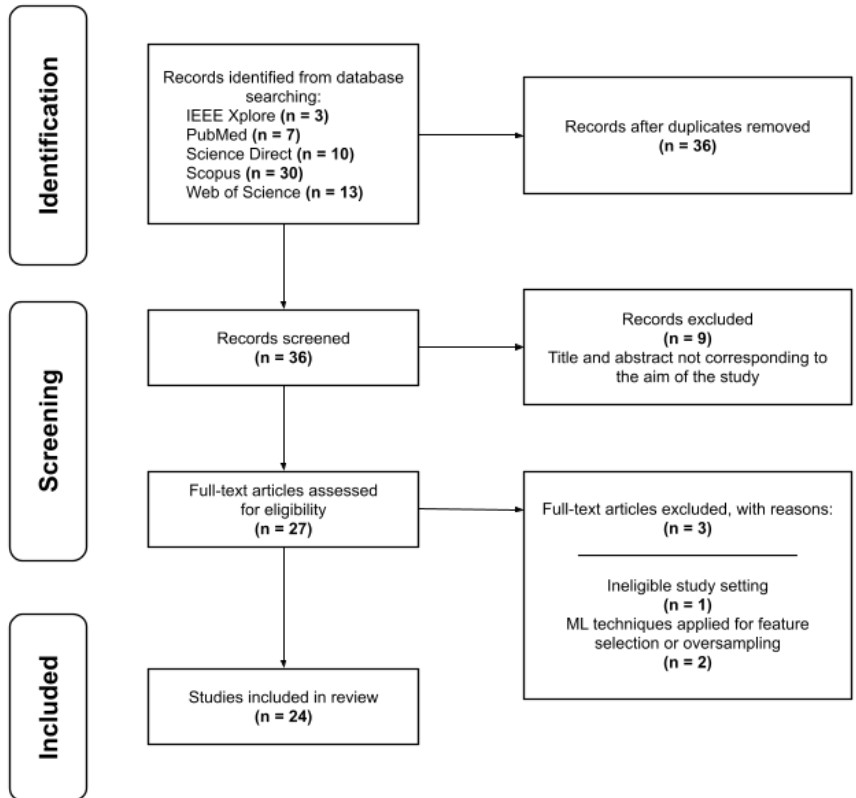

**Figure 1.** The literature selection process.

To describe the selected articles and the relations between them, as well as to develop knowledge that could not be understood by reading isolated studies, we analyzed and synthesized the data in order to extract some key information. We read the full papers (n = 24) and documented the following information:

1. Year of publication.
2. Dataset attributes.

        (a)      Number of patients
        (b)      Number of medical appointments
        (c)      Data collection period (year)
        (d)      Country

3.    No-show rate: percentage of absenteeism across the entire dataset.
4.    Medical specialty: area of medicine in which the study was applied.
5.    Algorithms used.
6.    Selected attributes: most important predictors that influence the behavior of the model.
7.    Model performance: evaluation metrics applied.

## 3. Results

The surveyed studies aim to explore machine learning techniques focusing on helping to understand and mitigate no-shows in medical appointments. Before starting to describe each of the studies, we illustrate the number of articles published over the years in Figure 2. It is noted that the number of published scientific articles referring to the no-show with the use of machine learning techniques is increasing nowadays. This fact is mainly due to the greater availability of data and the improvement of machine learning tools, such as the libraries available in different programming languages that encapsulate the complexity of implementing the algorithms.

It is worth mentioning that the problem of absenteeism in the health area has been the subject of scientific study for many years. However, most publications concern solutions focused on reducing no-shows through statistical techniques. With the advent of artificial intelligence in recent years, there was the possibility of using different machine learning techniques and optimizing solutions to mitigate no-show in medical appointments.

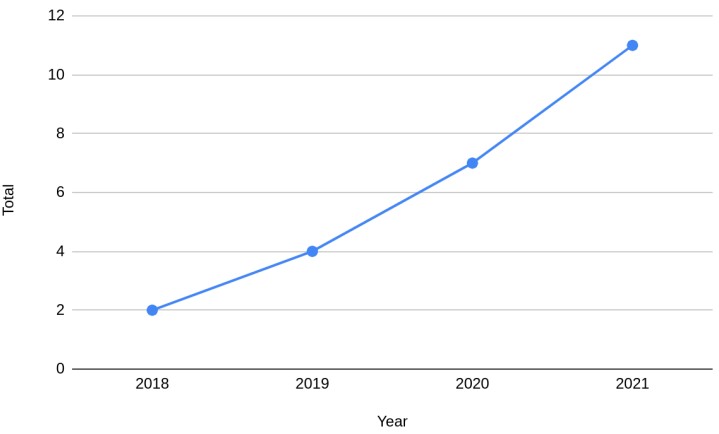

**Figure 2.** Number of articles published over the years.

Q1.   How much data is used to train machine learning models for no-show prediction? In total, 18 different data sets were used in the 24 studies found. Two studies [3,6] did not report the volume of data used. Only five studies used an amount of data greater than one million records, while the others used around 120 thousand records, on average. Seven works developed their solutions with the same dataset. Despite this, the works presented different approaches in the exploratory analysis of the data and in the algorithms used.

Regarding the dimensionality of the datasets, on average, 17 attributes were used to build the model in each work. The number of attributes for each work was counted based on the amount used to train the model, and not necessarily with the original attributes of the dataset. In this way, in most cases, not only the attributes of the original dataset were used, but new attributes were added through feature selection, for example.

The origin of the data was concentrated in nine different countries. Only the work by Qureshi et al. [10] did not describe the origin of the data used. Followed by the United States, Brazil was the country of origin of eight studies, seven of which used the same dataset. Regarding the data domain, that is, which context or specialty the data refer to, the vast majority concerns the primary care of patients or centers with care of multiple specialties.

With regard to the no-show rate present in the initial dataset, the highest rate observed was 85% [11]. Although this percentage represents a discrepant value compared to other works, the average absenteeism rate remained around 18% and the lowest rate was 10%. Of the 24, six works did not clearly describe the percentage of no-shows of the explored datasets. Table 5 summarizes the results in relation to the volume of data, in descending order with reference to the volume of data explored.

**Table 5.** Data Volume.

| Articles | Volume | No-Show Rate | No. of Features | Service | Country |
|----------|--------|--------------|-----------------|---------|---------|
| [11] | 33,050,363 | 85% | 29 | Primary Care | Saudi Arabia |
| [10] | 6,000,000 | - | 17 | Multiple Specialty | - |
| [12] | 2,362,850 | 10% | 12 | Multiple Specialty | Spain |
| [13] | 2,011,813 | 26.71% | 20 | Multiple Specialty | Saudi Arabia |
| [14] | 1,087,979 | 11.3% | 11 | Multiple Specialty | Saudi Arabia |
| [7] | 454,217 | 11.10% | 15 | Multiple Specialty | China |
| [5] | 374,072 | 26% | 26 | Primary Care | United States |
| [15] | 194,458 | - | 30 | Neurology | United States |
| [2,4,16–20] | 110,528 | 20.19% | 14 | Primary Care | Brazil |
| [21] | 101,534 | 11.39% | - | Pediatrics | Saudi Arabia |
| [22] | 76,285 | 30% | 18 | Family Medicine | United States |
| [23] | 53,311 | 21%–39% | 7 | Primary Care | Colombia |
| [8] | 32,957 | 17.40% | 21 | Radiology | Singapore |
| [24] | 25,523 | - | 16 | Cardiology | United States |
| [25] | 8,794 | 13.40% | 14 | Pediatrics | United States |
| [26] | 4,812 | - | 11 | Pediatrics | Brazil |
| [3] | - | - | 9 | Primary Care | United States |
| [6] | - | 8.60% | 20 | Multiple Specialty | Wales |

Q2. Which machine learning algorithms are used in the predictive models and which presented the best performance?

Table 6 summarizes the data found regarding the algorithms used in each work. They are presented in ascending chronological order and, for each work, the symbol "x" was inserted in the algorithms used and the symbol "o" for the algorithms used, and that obtained the best performance during the tests. The last row ("Total") shows the number of times the algorithm was used and obtained the best performance.

In general, the works used more than one machine learning algorithm to build the model, except for work [3]. Because the present work focuses on predictive models, most of the algorithms used in the works found were for classification, such as decision trees and logistic regression. However, regression algorithms such as linear regression and support vector regression (SVR) were also used in studies in which the objective

was also to predict the number of patients who would not attend consultations in a given time window [3,25].

**Table 6.** Machine learning algorithms. Symbols "x" and "o" represent tested and best performance algorithms, respectively.

| | AdaBoost | Artificial Neural Networks (ANN) | Bagging | Decision Tree (DT) | Deep Neural Networks (DNN) | Gradient Boosting (GB) | Gradient Boosting Machine (GBM) | General Linear Model (GLM) | Hoeffding Tree | JRip | K-Nearest Neighbors (KNN) | Linear Regression | LightGBM | Logistic Regression | Multilayer Perceptron (MLP) | Naive Bayes (NB) | Random Forest (RF) | Stochastic Gradient Descent (SGD) | Stacking | Support Vector Classifier (SVC) | Support Vector Machine (SVM) | Support Vector Regression (SVR) | XGBoost |
|---|---|---|---|---|---|---|---|---|---|---|---|---|---|---|---|---|---|---|---|---|---|---|---|
| [24] | | | | | x | | | | | | | | | | | | x | o | | | | | |
| [26] | | | | x | | | | | | | | | | x | | | | o | | | | | |
| [10] | x | | | x | | x | | | | | | | | x | x | x | o | x | | | x | | x |
| [21] | | | | | | | o | o | | | | | | o | | | | | | | | | |
| [5] | o | | o | x | | o | | | | | | | | x | | | o | | | | | | |
| [6] | | | | | | | | | | | | | o | x | | | | | | | | | |
| [7] | | | o | x | | x | | | | | x | | | x | | | x | | | | | | |
| [2] | | | | o | | | | | | | x | | | | | x | | | | | x | | |
| [16] | x | | | o | | | | | | | | | | | | | | | | | | | |
| [17] | | x | | | | o | | | | | | | | x | | | x | | | | | | |
| [3] | | | | | | | | | | o | | | | | | | | | | | | | |
| [18] | | | | o | | | | | | | | | | x | | | x | | | | | | |
| [4] | | x | | | | | | | | | | | | x | | | o | | | | x | | |
| [13] | | | | | | o | | | | | | | | x | x | | x | | | | x | | |
| [8] | | x | | | | | | | | | | | | x | | | x | | | | | | o |
| [25] | | | | | | | | | | | | | | | | | x | | | | | x | o |
| [23] | | o | | | | | | | | | | | | x | | x | | | | | | | |
| [11] | x | | | | o | | | | | | | | | | | | x | | | | | | |
| [19] | | | | | | o | | | | | x | | | x | | | x | | | | | | |
| [20] | | | | | | | | | | | x | | | x | | | x | o | | o | | | |
| [14] | | | | | | | | | o | x | | | | | | | | | | | | | |
| [15] | | | | x | | | | | | | | | | | | x | | o | | | | | |
| [22] | | x | | | | | | | | | | | | x | | | x | x | | o | | | |
| [12] | | x | | | | o | x | | | | | | | | | | | | | | | | |
| Total | 1 | 1 | 2 | 3 | 1 | 4 | 1 | - | 2 | 1 | - | 1 | 1 | 1 | - | - | 5 | 2 | 1 | 1 | - | - | 2 |

Decision trees, logistic regression, and random forest are classic algorithms that have been included in most works. Due to recent studies showing that ensemble algorithms can improve the performance of classical algorithms, some works have included them in the tests. For example, the work by Lekham et al. [5] used the bagging, AdaBoost, and gradient boosting algorithms and obtained good performance among all the candidate algorithms.

The algorithms with the best performance in the works found were Gradient Boosting, Decision Trees, and Random Forest. Even if several complex algorithms as the ones based in Artificial Neural Networks (ANN) were trained to build the models, besides performance, the model interpretability was considered to select the candidate models in the articles. Random forest was selected as the most suitable for building the models in most works. The performances of the algorithms of each work are presented in Table 7. The metrics used in the works focus on the Area Under the ROC Curve (AUC-ROC) as it is a metric that clearly describes the results obtained from the model.

**Table 7.** Algorithms performance.

| Articles | Algorithms | Performance |
|---|---|---|
| [24] | Stochastic Gradient Descent | 0.85 (AUC-ROC) |
| [26] | Random Forest | 0.969 (AUC-ROC) |
| [10] | Random Forest | 0.9209 (AUC) |
| [21] | Logistic regression, JRip and Hoeffding Tree | 0.86 (F-Score) |
| [5] | Random Forest, Ada Boost, Gradient Boosting and Bagging | 0,73 (F-Score) |
| [6] | LightGBM | 0.37 (F1-Score) |
| [7] | Bagging | 0.990 (AUC) |
| [2] | Decision Tree | 95% accuracy |
| [16] | Decision Tree | 0.88 (AUC-ROC) |
| [17] | Gradient Boosting | - |
| [3] | Linear Regression | 0.72 (AUC-ROC) |
| [18] | Decision Tree | 78-80% accuracy |
| [4] | Random Forest | 0.8678 (AUC-ROC) |
| [13] | Gradient Boosting | 0.81 (AUC-ROC) |
| [8] | XGBoost | 0.746 (AUC-ROC) |
| [25] | XGBoost | 0.90 (AUC-ROC) |
| [23] | Artifical Neural Network | 0.90 (AUROC) |
| [11] | Deep Neural Networks | 0.982 (precision), 0.943 (recall) |
| [19] | Gradient Boosting | 80.91% accuracy |
| [20] | Support Vector Classifier and Stochastic Gradient Descent | 0.68 (AUC-ROC) |
| [14] | Hoeffding Tree | 77.13% accuracy |
| [15] | Random Forest | 0.68 (AUC) |
| [22] | Stacking | 0.846 (AUC) |
| [12] | Gradient Boosting Machine | 0.7404 (AUC-ROC) |

Q3. What characteristics most influence patients not to attend a scheduled medical appointment?

The attributes of the datasets explored in the works represent the characteristics related to the patients demographics, the appointment, and the patients' behavior during the appointment. Most of the works found created new attributes based on the existing ones or added new datasets, in order to have more characteristics of the problem represented through the data. The lead time attribute was created in all works, where the appointment and appointment date information was available. This attribute represents the distance from the day of appointment scheduling to the appointment day and has a considerable impact on the models.

Except for three works [11,20,25], all others consider that the most influential factors in the built model are related to the patient's age, whether the patient missed a previous appointment (previous no-show), and the distance between the appointment and the patient's scheduling (lead time). Other attributes, such as the geographical distance from the patient's home to the clinic location, appointment date and shift, medical specialty, and whether there was prior confirmation of the appointment, had a low impact on the algorithms. Table 8 presents a summary of this information.

**Table 8.** Feature importance.

| | Patient Demographics | | Appointment | | | | | Patient Behavior | |
|---|---|---|---|---|---|---|---|---|---|
| Articles | Age | Distance | Month | Shift | Confirmation | Type | Specialty | Previous No-Show (%) | Lead Time |
| [24] | | | | | | | | X | X |
| [26] | | | X | | X | | | | |
| [10] | X | | | | | X | | | X |
| [21] | X | X | X | | | | X | | |
| [5] | X | | | | | | | | X |
| [6] | X | | | | | | X | | X |
| [7] | X | X | | | | X | | | |
| [2] | | | | | | | | | X |
| [16] | | | | X | X | | | | X |
| [17] | | X | | | | | | | X |
| [3] | X | | | | | X | | X | |
| [18] | X | | | | X | | | | |
| [4] | X | | | | | | | X | X |
| [13] | | | | | | | | X | X |
| [8] | X | | | | | | | | X |
| [25] | | | | | | | | | |
| [23] | | | X | | X | | | | X |
| [11] | | | | | | | | | |
| [19] | X | | | | | | | | X |
| [20] | | | | | | | | | |
| [14] | | | | | | | X | X | |
| [15] | | X | | | | X | | X | X |
| [22] | | | X | X | | | | | X |
| [12] | X | X | | | | | X | | |
| Total | 11 | 5 | 4 | 2 | 4 | 4 | 4 | 6 | 14 |

Q4. What is the no-show rate reduction in medical appointments that solutions developed with the help of machine learning techniques achieve?

Only the work by Chong et. al [8] described the experiment performed in which there was a reduction in the number of no-show appointments in practice. During the six months of the experiment, there was a reduction from 19.3% to 15.9% in abstentions,

by sending a reminder to 25% of the patients who were pointed out by the model as at greater risk of not attending the appointment. The other works did not present metrics that demonstrate the reduction of no-shows in medical appointments, after the machine learning model development.

Four works [10,18,20,26] only presented the exploratory analysis of the data and the steps for the model building. However, they did not implement or discuss software and/or process management solutions that could be developed based on the study. Prior communication with patients through electronic reminders, such as SMS or phone calls, was presented as a solution in most studies, as in [5–7]. In order to avoid the high cost of sending reminder notifications to every patient, some studies have proposed to optimize the sending and make them only for patients who are at high risk of no-show [2,7,16].

The study by Srinivas and Salah [24] proposes, as an alternative to reduce absenteeism in medical appointments, the subsidy for the patient's transport to the health center, regarding the economic profile of the patients. Other studies [2–4] propose the use of overbooking techniques as a way of mitigating the patients' non-attendance.

The optimization of appointment planning through the integration of the predictive model developed in the scheduling system [3,5,16,21], the collection of basic information for new patients through an online form [24], and the remodeling of workflow and scheduling policies [21] are alternatives mentioned in the studies as a way of reducing absenteeism.

## 4. Discussion

The results revealed that the solutions used to mitigate medical appointments no-show with machine learning techniques do not present practical results of reducing abstentions based on the solutions presented. Except for the work by Chong et al. [8], which showed in their experiments that there was a 3.4% reduction in no-shows after the deployed solution.

Although some works have explored a dataset with a large volume of data, it was observed that the models are not directly impacted by this. The accuracy of the models developed with a smaller amount of data was higher or equivalent to the others. For example, Alshammari, Daghistani and Alshammari [11] built the model with a dataset four orders of magnitude larger in the number of records compared to [26], and the performance of the algorithms was equivalent.

The data quality is evidenced as an important factor in terms of building the machine learning models. Except for a few works, most have derived new attributes from existing ones from the original dataset or added new datasets to improve the model's comprehensiveness. Similarly, some re-balancing techniques were performed in most of the datasets in order to not bias the model training. Although the focus and objective of the research carried out and the results obtained have no focus and relationship with the treatment of patient data privacy, we use definitions, characteristics and criteria defined in these works [27–30]. These works present research and useful contributions to data privacy management in different concepts and scenarios.

Except for three works [11,20,25], all the others presented the most important features for the given model based in a feature importance measure. Depending on the dataset with which the algorithms were trained, different attributes were highlighted as the features that have a larger effect on the model. Although, the age of the patient, prior no-show history and the lead time were pointed out by the models as the features that mostly affected the no-show rate.

The results show that the most appropriate algorithms for building the models are the decision tree algorithms, either using a binary decision tree or using ensemble methods, such as Random Forest. The work by Qureshi et al. [10] tested several more complex algorithms for building the model, such as the Multilayer Perceptron (MLP), but obtained satisfactory performance with the Random Forest algorithm. The choice of the final algorithm was also guided by the complexity of understanding the model built, such as

the choice of decision trees against algorithms considered a black box, such as artificial neural networks.

## 5. Conclusions

This work presents the state-of-the-art regarding the use of machine learning techniques to mitigate no-show in medical appointments. The use of relevant terms and inclusion and exclusion criteria allowed the mapping of the most relevant scientific studies published. The selected works helped to identify (i) the volume of data used for training the models, (ii) which algorithms are used, (iii) which characteristics most influence the no-show behavior of patients, and (iv) what each no-show reduction rate presented by the solutions found.

In fact, the relevance of the problem can be observed based on the 24 articles analyzed. This review has identified patient characteristics that were most frequently associated with no-show behavior: age of the patients, whether the patient missed a previous appointment, and the lead time between the appointment and the patient's scheduling. Additionally, the results indicate that even with more complex algorithms available in artificial intelligence, decision trees algorithms are still the best choice to handle the no-show in medical appointments.

As is shown in surveyed studies, the fact that no-shows in medical appointments do not occur randomly, predicting the risk for individual patients not attending can help to mitigate the impact on health centers. Understanding the patient's no-show behavior through analysis of different health contexts and with a major variety of features is a key problem new studies should address.

Our findings show that solutions developed with the help of machine learning techniques are still not widely integrated in scheduling systems. The majority of the surveyed studies discuss solutions to mitigate no-shows with the help of machine learning, but only one work has validated the experiment in a real world environment. Therefore, this study is useful for medical center administrators interested in applying changes in their scheduling systems based on artificial intelligence solutions, and also for researchers who seek to explore literature dealing with no-show appointments and machine learning algorithms.

**Author Contributions:** Conceptualization, L.H.A.S., W.D.P., and A.M.d.R.F.; methodology, L.H.A.S., W.D.P., and A.M.d.R.F.; analysis, L.H.A.S., W.D.P. and A.M.d.R.F.; writing—original draft preparation, L.H.A.S., V.R.Q.L., W.D.P. and A.M.d.R.F.; writing—review and editing, L.H.A.S., V.R.Q.L., W.D.P., and A.M.d.R.F. All authors have read and agreed to the published version of the manuscript.

**Funding:** This research was funded in part by the Coordenação de Aperfeiçoamento de Pessoal de Nível Superior, Brasil (CAPES)—Finance Code 001 and Fundação de Amparo à Pesquisa do Estado de Santa Catarina (FAPESC)—EDITAL DE CHAMADA PÚBLICA FAPESC No. 06/2017.

**Institutional Review Board Statement:** Not applicable.

**Informed Consent Statement:** Not applicable.

**Data Availability Statement:** Not applicable.

**Acknowledgments:** Fundação para a Ciência e a Tecnologia, I.P. (Portuguese Foundation for Science and Technology) by the project UIDB/05064/2020 (VALORIZA–Research Centre for Endogenous Resource 764 Valorization), and Project UIDB/04111/2020, ILIND–Instituto Lusófono de Investigação e Desenvolvimento, under project COFAC/ILIND/ COPELABS/3/2020.

**Conflicts of Interest:** The authors declare no conflict of interest.

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
