# Peer review of "No-Show in Medical Appointments with Machine Learning Techniques: A Systematic Literature Review"

_information, doi:10.3390/info13110507_

Round 1

Reviewer 1 Report

In this paper, the authors have presented a literature review on the use of machine learning techniques to mitigate no-show in medical appointments. The paper is well-written, but I have the following comments.

1.     The research questions are quite interesting but the author’s need to provide proper motivation behind selection of the research questions.

2.     Why was the literature survey limited to the last 5 years? Why was not older papers considered for this study?

3.     Before presenting the results, it will be interesting to have a table of the papers with couple of bullet points mentioning the most important result of each paper. This is just a suggestion.

4.     The authors should provide a discussion that what are most important parameters that the machine learning models predict that mostly affected the no-show rate.

5.     The authors should provide a summary of the future direction that they could find from the papers,

Author Response

Dear Reviewer

Thank you for your time and availability in reviewing our work, attached we sending you a letter detailing the changes made.

Reviewer 2 Report

In this paper,  a systematic review of machine learning techniques applied to no-show appointments is discussed. The paper can be a good contribution provided the authors can revise based on the following recommendations. 

1. The objective of the study should be clearly emphasized. 

2. Based on what parameters and features the study was conducted.

3. From which year to which year papers are considered?

4. Is this the first work in this area? or other works also present. If it exists what improvements in this work need to be clarified?

5. Apart from the decision tree, what other algorithms were considered in past studies? Why they were not that much approved?

6. Some more analysis based on different features needs to be considered.

7. Reference can be improved by considering recent works.

Author Response

(The authors gave the same response as above.)

Round 2

Reviewer 1 Report

The authors have successfully addressed the comments. The quality of the paper has improved after the revision and I recommend it for publication.